# Structure, Substrate Specificity and Role of Lon Protease in Bacterial Pathogenesis and Survival

**DOI:** 10.3390/ijms24043422

**Published:** 2023-02-08

**Authors:** Perumalraja Kirthika, Khristine Kaith Sison Lloren, Vijayakumar Jawalagatti, John Hwa Lee

**Affiliations:** 1Department of Biochemistry and Molecular Biology, Mayo Clinic, Rochester, MN 55902, USA; 2Department of Urology, Mayo Clinic, Rochester, MN 55902, USA; 3College of Veterinary Medicine, Jeonbuk National University, Iksan 54596, Republic of Korea

**Keywords:** Lon protease, bacteria, virulence, stress response, pathogenesis

## Abstract

Proteases are the group of enzymes that carry out proteolysis in all forms of life and play an essential role in cell survival. By acting on specific functional proteins, proteases affect the transcriptional and post-translational pathways in a cell. Lon, FtsH, HslVU and the Clp family are among the ATP-dependent proteases responsible for intracellular proteolysis in bacteria. In bacteria, Lon protease acts as a global regulator, governs an array of important functions such as DNA replication and repair, virulence factors, stress response and biofilm formation, among others. Moreover, Lon is involved in the regulation of bacterial metabolism and toxin–antitoxin systems. Hence, understanding the contribution and mechanisms of Lon as a global regulator in bacterial pathogenesis is crucial. In this review, we discuss the structure and substrate specificity of the bacterial Lon protease, as well as its ability to regulate bacterial pathogenesis.

## 1. Introduction

A dedicated group of proteases carry out intracellular proteolysis, which is essential for cell survival. The proteases play a pivotal role in maintaining cellular protein homeostasis by removing damaged, non-functional and short-lived proteins, especially those under stress conditions that threaten the proteome. By regulating the amount of specific functional proteins, the proteases participate in transcriptional and post-translational pathways. In bacteria, intracellular proteolysis is carried out by ATP-dependent proteases that belong to the large AAA+ family proteins (ATPase associated with various cellular activities), such as Lon, FtsH, HslVU and the Clp family (ClpAP and ClpXP) [1,2]. Among them, FtsH is the only essential and membrane-anchored protease [3], and Lon is present in all of the kingdoms of life. Lon dominates proteolysis in the cytosols of bacteria, the plasma membranes of archaea and the mitochondria of eukaryotes [4]. In *Escherichia coli*, more than half of the intracellular proteolysis is carried out by Lon protease [5]. The bacterial Lon protease (LonA) consists of three major domains: the N-terminal domain (NTD) that oligomerizes and recognizes substrates, the hexameric AAA+ (A) domain with unfolding activity, i.e., ATP binding and hydrolysis, and the C-terminal serine protease (P) domain, which hydrolyses substrates [6,7]. In the absence of a substrate, the enzyme’s hexameric ring adopts an open conformation. On the other hand, in the closed substrate-engaged form, sequential ATP hydrolysis in the adjoining subunits around the ring drives the translocation of unfolded substrates from the A domain towards the P domain via the of them binding to a staircase of aromatic residues [8,9]. This mechanism is most likely to be conserved in all Lon proteases and is related to the rotary treadmilling mechanism of other AAA+ translocases [10]. Cryo-EM structure studies of the human counterpart called LONP1 protease, which is found in the mitochondria, revealed the structural conservation between human and bacterial Lon protease, which is suggestive of similar mechanisms of action [11,12]. Because of their important cellular functions, human Lon proteases are considered to be promising therapeutic targets [13], and their corresponding bacterial homologs are considered to be potential antimicrobial drug targets [14]. Hence, advancing the knowledge of the substrate pools of specific proteases and the mechanisms underlying substrate selection is vital.

Lon protease-mediated proteolysis regulates various functions in bacteria including DNA replication and repair [15,16], the upregulation of virulence factors [17,18,19,20], persister formation, encapsulation [7], motility [21], the SOS response [22], UV sensitivity [21], biofilm formation [23], sporulation [24] and heat shock and stress responses [25,26,27,28]. Studies have shown that Lon protease plays a pivotal role in withstanding stress in *Francisella tularensis* [29], *Vibrio cholerae* [19] and *Dickeya solani* [30]. Studies of *Escherichia coli* have showed that the levels of Lon-specific mRNA were upregulated after exposure to salt and oxidative stresses or after a treatment with puromycin [31]. Further, Lon proteolytical regulation contributes to bacterial adhesion and invasion by modulating the host cytoskeleton in *Salmonella* Typhimurium [18]. Moreover, Lon is involved in the regulation of bacterial metabolism and toxin–antitoxin systems [32]. In the literature, there are several studies that show that the overexpression of Lon protease can be either beneficial or detrimental depending on the organism [33,34]. In *Streptomyces coelicolor*, an additional copy of the Lon protease gene increases the rate of antibiotic synthesis [35], whereas in *Bacillus thuringiensis*, the overexpression of Lon leads to the degradation of activator proteins that are vital for motility, thereby limiting bacterial motility [2] (Figure 1).

In the current review, we discuss the structure of the bacterial Lon protease, its substrate specificity, and its ability to regulate proteolysis in bacteria. Further, we summarize the role of Lon in regulating virulence factors, survival in stress conditions and modulating the bacterial pathogenesis of the host cells.

## 2. Characterization of Lon Protease

### 2.1. Structure of Bacterial Lon Protease

Lon proteases assemble into barrel-shaped homo-hexamers, with the proteolytic active sites sequestered in an internal chamber, and they are largely inaccessible to folded proteins; this architecture serves to prevent the degradation of non-substrate proteins [9]. A single monomer of Lon contains an ATPase AAA+ molecular chaperon and a proteolytic domain with a serine-lysine catalytic dyad. Additionally, the bacterial Lon proteases (LonA) also contain an extra two-domain N-terminal region, with all of the domains being part of a single polypeptide chain. The N-domain is essential for the oligomerization of LonA proteases and is involved in the binding of protein substrates and their hydrolysis. In *Escherichia coli*, the enzyme assembles into a hexamer with an internal degradation chamber accessible via an axial pore in the AAA+ ring [36]. The Lon hexameric ring recognizes a substrate, unfolds the protein if it is necessary by ATP-dependent reactions mediated by the AAA+ pore and translocates the denatured polypeptide through a central axial pore and into the proteolytic chamber for degradation [36,37] (Figure 2).

Apart from the well-known hexameric form of Lon protease, larger oligomeric forms of the enzyme are reported in *Escherichia coli*, *Bacillus subtilis* and *Mycobacterium smegmatis* [36,39,40]. In *Escherichia coli*, Lon forms dodecamers that equilibrate with hexamers at a physiological concentration. Here, the N-domain interactions between the Lon hexamers lead to the formation of a Lon dodecamer with an altered, yet unique, substrate with degradation properties. For the protein substrates to gain access to the degradation machinery in the dodecamer, they need to pass through the equatorial portals. By doing so, the dodecamers prevent proteolysis of the large, but not small protein substrates. This selective protein degradation helps to maintain the intracellular repertoire of the Lon substrate protein [36]. An unexpected dimeric interaction between the N-terminal domains and the Lon monomers is thought to initiate a possible dodecamer formation in *Bacillus subtilis*. Here, a domain swap arrangement in the N-terminal domain dimers may facilitate the head-to-head dimerization of hexamers, which might help in the formation of a dodecamer [39]. Ion-dependent oligomerization is essential for Lon protease activity in *Mycobacterium smegmatis*. At low Mg^2+^ concentrations, the cellular conditions favor the formation of lower oligomeric forms of Lon, whereas at higher Mg^2+^ concentrations, the enzyme exists in higher order oligomers [41].

### 2.2. Substrate Specificity of Bacterial Lon Protease

Even though the exact role of the three Lon domains in substrate selection is unknown, the N-terminal domain of the *Escherichia coli* Lon is essential for substrate recognition and binding, as shown by the direct binding between Lon and sul20 peptide [7,42]. Concomitantly, the literature shows that an E240K mutation in the N-terminal domain alters the degradation of substrates, and a truncation of the N-terminal domain completely impairs the *Escherichia coli* Lon proteolytic activity towards β-casein [43]. The Lon proteases are renowned for their ability to degrade misfolded and faulty proteins, which is an observation that has been substantiated by studies carried out in *Escherichia coli*. Yet, a few proteins undergo Lon-mediated proteolysis even in their native forms, namely SulA [44], SoxS [45], HupB [46], IbpA [27], RcsA [47] and RpsB [48]. We lack elucidations that show how protein substrates are earmarked for degradation by Lon protease. Certain sequence features on the known substrates seem to contribute to their degradation by Lon. In this regard, Lon is similar to other AAA+ proteases that recognize specific peptide signals in substrates called degradation tags or degrons [49]. Attaching a degradation tag to a stably folded protein leads to its lysis by Lon [50]. In *Escherichia coli*, a degron SsrA tag (AANDENYALAA) marks the polypeptides for targeted proteolysis by Lon [51]. Additionally, the Lon-mediated degradation of certain substrates requires adaptors. This is exemplified by the observation that *Bacillus subtilis* SmiA facilitates the degradation of SwrA by Lon [26]. Furthermore, studies of *Yersinia pestis* have showed that HspQ acts as a Lon specificity-enhancing factor and enhances the Lon-mediated degradation of a select set of substrates including YmoA, RsuA, Y0390 and Fur [52]. Karlowicz et al. reported the degradation of certain DNA-binding proteins, namely TrfA and RepE, which require the interaction of Lon protease with DNA [53]. The important bacterial proteins that are substrates of Lon protease are stated in Table 1. Another distinct feature of Lon protease is its organism specific substrate specificity. A recent study showed that the human Lon protease failed to degrade *Francisella tularensis* and *Escherichia coli* Lon substrates. Furthermore, a domain swap between the bacterial and human Lon proteases showed that substrate recognition and biochemical cleavage require there to be an intrinsic match of three domains in the Lon proteases [7]. It is not clear if processive peptide bond hydrolysis without substrate dissociation is necessary for Lon-mediated degradation. The studies across species have shown that Lon proteases degrade the substrate and generate peptide products consisting of ~5–30 amino acids [54,55].

### 2.3. Role of Redox Switch in Modulating Lon-MEDIATED Proteolysis

The Enterobacteriaceae family members, e.g., *Escherichia coli*, *Shigella*, *Salmonella*, *Serratia* and *Yersinia pestis* are mostly symbionts dwelling in the gut of the host [71]. These bacteria are some of the commonly encountered pathogens in clinical microbiology and cause a wide range of diseases. They are facultative anaerobes that live in both harsh aerobic and anaerobic environments. The Lon protease redox switch seems to play a vital role in enabling the facultative lifestyle of these organisms. Nishii et al. [4] reported that the conserved cysteine residues in Lon proteases of these bacteria act as a redox switch, which changes the size of the exit pore of the P-domain ring to regulate the proteolytic activity depending on the availability of oxygen. In *Escherichia coli* Lon, only Cys617 and Cys691 are sensitive to the redox conditions, whereas the other cysteine residues seem to be impervious to it. In an aerobic environment, the microbes constantly encounter oxygen and oxidants, as well as reactive oxygen species that are produced as byproducts of aerobic respiration. This oxidative stress leads to the production and accumulation of damaged and misfolded proteins. The high activity level of the oxidized form of Lon protease helps to eliminate the intracellular faulty proteins. Although the high activity level of Lon may pose a threat to the native cellular proteins, their replenishment is swift in the oxygen-rich environment, as aerobic respiration rapidly produces the ATP required for protein synthesis. On the other hand, upon entering the host’s body, these bacteria live in the intestine, and primarily, in the colon, the regions that are almost free of oxygen. These anaerobic environments slow the rate of cellular protein synthesis due to inefficient production of ATP, thereby reducing the amount of improperly synthesized and/or folded proteins. Thus, the low activity level of the reduced form of Lon protease is suitable for anaerobic environments. Consequently, these pathogenic bacteria can fine-tune the activity of Lon protease by an extraordinary mechanism to live robustly both inside and outside the host body [4].

## 3. Role of Lon Protease in the Virulence of Pathogenic Bacteria

Lon protease is well known to be involved in housekeeping protein turnover, as well as in the regulatory proteolysis [72]. Studies have shown the role of Lon protease in regulating virulence-associated genes in several bacterial pathogens [73,74,75] (Table 2). Here, we broadly describe how Lon modulates the virulence factors in different disease-causing bacteria.

### 3.1. Pseudomonas Aeruginosa

This Gram-negative bacterium is the third most common cause of nosocomial infection. It causes urinary tract infections, pneumonia and death in patients suffering from cystic fibrosis (CF) due to chronic infections, leading to irreversible lung damage and respiratory failure. In recent years, Pseudomonas has developed a resistance to common antibiotics through adaptations and mutations, thereby making it difficult to treat the infection in patients [75,76]. Lon plays a vital role during the early stages of infection, wherein it is essential for the bacterial invasion of the respiratory tract, as well as adherence to the epithelium. Additionally, Lon is essential for the establishment of chronic infection as it regulates the bacterial colonization in the lungs. The inactivation of Lon leads to a decrease in the cytotoxicity of the host’s cells. This observation was substantiated by the downregulation of the expression of Type III secretion genes including *pscF* (needle complex), *popB* and *popD* (translocation apparatus), *exsC* (regulatory protein) and *exoS* (effector protein cytotoxin). The Lon-mediated dysregulation of the Type III secretion system is crucial to inhibit phagocytosis by the host’s neutrophils. Further, *Pseudomonas aeruginosa* Lon modulates cell division, promotes surface motility and helps the bacteria to develop resistance against ciprofloxacin [75].

### 3.2. Salmonella Typhimurium

*Salmonella enterica* serovar Typhimurium is a facultative intracellular bacterial pathogen, causing diseases ranging from localized gastroenteritis to severe, life-threatening systemic diseases [77,78]. The role of Lon protease in Salmonella pathogenesis has been extensively studied and reported. Upon the deletion of Lon, *Salmonella* Typhimurium loses its ability to cause systemic diseases in mice. Lon negatively regulates the genes that are essential for the adhesion and invasion of *Salmonella* Typhimurium in both phagocytic and non-phagocytic cells [18]. The expression of invasion genes found in *Salmonella* pathogenicity island 1 (SPI-1) is important for the penetration of the intestinal epithelium. The inactivation of lon upregulates the transcriptional levels of SPI1 genes *hilA*, *invF*, *sipA* and *sipC*, thereby promoting bacterial invasion into the host’s epithelium [1,79]. Lon downregulates the SPI-1 genes through the degradation of the regulator proteins HilC and HilD [80]. On the other hand, Lon deletion represses the expression of SPI-2 genes. This observation corroborates the study of genes encoding the SPI-2 effectors PipA and PipB, which were significantly downregulated [20]. Additionally, Lon downregulates SPI-4, SPI-9 and flagellar genes, thereby facilitating intracellular bacterial replication in the macrophages. This suggests that Lon may oppositely modulate two major infection stages of *Salmonella* Typhimurium: epithelial invasion and survival within the phagocytes.

### 3.3. Yersinia Species

Members of the enteropathogenic Yersiniae family (*Yersinia enterocolitica* and *Yersinia pseudotuberculosis*) rely on an array of virulence factors to adhere, enter and colonize the host, evade the host’s immune response, and ultimately, trigger the host cell’s suicide response [81]. To escape the host, evade the innate immune system and replicate and disseminate extracellularly, *Yersinia* species are equipped with an approximately 70 kb plasmid, which is called pCD1 in *Yersinia pestis*. Plasmid pCD1 and related plasmids in the Yersiniae encode a set of secreted anti-host proteins that are termed *Yersinia* outer proteins (Yops) and a delivery system classified as a type III secretion system (T3SS). The transcription of genes encoding Yops and T3SS are activated by temperatures of 37 °C in the presence of millimolar concentrations of calcium, which are conditions that mimic the mammalian host. T3SS allows the bacteria to directly inject Yops into the host cell’s cytoplasm. Once they are inside, the injected Yops impede phagocytosis and block the production of pro-inflammatory cytokines. Additionally, they subvert the host cell’s signaling pathways and trigger a pre-programmed metabolic chain reaction that results in the host cell’s suicide response [82,83]. A small histone-like protein called YmoA has been shown to have repressor-like activity on pCD1 operon expression, which means that it negatively regulates the expression of T3SS. Further, the inactivation of Lon caused the accumulation of YmoA in *Yersinia pestis*. The inability of Lon mutants to secrete Yops and transcriptionally upregulate the T3SS substantiates that Lon protease regulates the proteolysis of YmoA, which in turn modulates the thermoregulated activation of T3SS-related operon in Yersiniae [68].

### 3.4. Brucella abortus

*Brucella* pathogenesis depends on its ability to replicate in various mammalian cells [84,85]. During this process, the bacteria alter the expression of many of their genes to adapt to the harsh environment in the host and maneuver through the immune system. Upon the inactivation of Lon, the transcriptional levels of TNF-α were reported to be upregulated in the macrophages. This observation elucidates that the Lon may be involved in TNF-α mediated cell death though an apoptotic or necrotic pathway in the host [86]. Additionally, the *Brucella* mutants deleted by Lon downregulated the mRNA levels of IL-2, IL-10 and interferon-γ and reduced the levels of mRNA for IL-4 in the murine splenocytes [87]. It is speculated that the Lon mediated-splenic swelling, as well as the integrity of the bacterial envelope, are essential to regulate the host’s cytokine response [88]. Lon regulates iron metabolism, thereby helping *Brucella* adapt to the intracellular iron-deficient environment and modulates the urease operon to enable the bacteria to tolerate acid stress [88]. The works of Robertson et al. on *Brucella abortus* Lon elucidated that this stress response protease is essential for bacterial virulence during the initial stages of the infection in the mouse model. On the contrary, the contribution of Lon for the establishment and maintenance of a chronic infection in the aforementioned animal model was reported to be non-essential [89].

### 3.5. Escherichia coli

The role of Lon protease in intracellular protein turnover and virulence is well elucidated in *Escherichia coli*. The enterohemorrhagic *Escherichia coli* (EHEC) virulence factors encoded in the locus of the enterocyte effacement (LEE) pathogenicity island are modulated by Lon protease. The inactivation of Lon in EHEC upregulates the expression of the *ler*, *grlA*, *espA* and *nleA* genes, signifying that Lon protease downregulates the activity of these virulence-associated genes. Additionally, Lon negatively regulates T3SS by modulating the expression of lipoprotein NlpE, a Cpx sensor that serves as a sentinel for protein sorting and folding defects in the *Escherichia coli* envelope [90,91]. Recent studies have shown that Lon protease in *Escherichia coli* reduces the rate of survival upon an antibiotic challenge [92]. Lon enhances the persister populations in *Escherichia coli* by upregulating their resistance to antibiotics such as ciprofloxacin [93,94]. Additionally, the inactivation of Lon helps *Escherichia coli* to tolerate higher concentrations of several antibiotics including tetracycline, kanamycin and erythromycin by upregulating the levels of MarA, an activator of drug efflux that functions through the AcrAB-TolC pump [95,96,97]. In addition to altering the bacterial ability to circumvent antibiotic stress, Lon plays a crucial role in modulating antimicrobial resistance in *Escherichia coli*. The deletion of Lon helps the bacteria to develop multidrug resistance faster than the wild type can by the insertion (IS) element-mediated duplication of genes coding for the AcrAB drug efflux pump [97]. Further, Lon deficiency can enhance the bacterial fitness and the potential for trimethoprim resistance due to mutations in dihydrofolate reductase (DHFR) enzyme [92,98].

## 4. Bacterial Lon Protease-Mediated Stress Response

Bacteria are equipped with an arsenal of stress response mechanisms that help them to cope with various stresses to survive in hostile environments both inside and outside of the host body [99]. Additionally, the bacteria have developed coordinated responses and perform crosstalk, which help the microorganism to grow and proliferate. Metabolic stress in the form of nutritional deprivation differentially inhibits overall mRNA translation and protein synthesis by a stringent response [100,101]. The inactivation of Lon in various species results in an inability to respond to stress conditions [32]. The role of ATP-dependent proteases in oxidative and heat shock stress responses are well elucidated [102,103]. Here, we describe the role of Lon protease-mediated stress response in different pathogenic bacteria.

### 4.1. Lon Protease Provides Bacteria with the Tool to Combat Stress Conditions

In a previously published study, we observed that the Lon is crucial in mitigating oxidative stress, which helps the bacteria to maneuver through the host’s phagosomes. The loss of Lon in *Salmonella* Typhimurium caused an increased accumulation of endogenous Hydroxyl radicals (OH·). To circumvent oxidative stress, Lon mutants upregulate the expression of outer membrane proteins to secrete H_2_O_2_ and prevent further damage to the cell’s proteome [18]. Thus, a well-coordinated network of proteins and pathways is regulated by Lon to prevent *Salmonella* Typhimurium from succumbing to free radical-mediated stress. Additionally, Lon is important in other abiotic stress (cold, acidic and osmotic ones) responses in *Salmonella* Typhimurium [18]. These findings were corroborated by an observation reported in other bacterial species. Exposure to a high temperature stimulates the expression of lon in *Escherichia coli* [104] and *Francisella tularensis* LSV [29]. The expression of *Escherichia coli* lon depends on the sigma32 transcription factor, which is activated under heat and osmotic stresses [31]. Ionic osmotic stress stimulates the upregulation of Lon expression in *Bacillus subtilis* and the phytopathogen *Dickeya dadantii* [105].

### 4.2. Lon Protease Modulates Toxin-Antitoxin (TA) Module Proteome

The TA modules consist of two genes organized on a single operon, the first one encoding an unstable antitoxin degraded by an ATP-dependent protease, and the second one encoding a stable toxin. The antitoxin counteracts the lethal activity of the toxin by direct protein–protein interaction. The TA systems are stress response modules that inhibit physiological functions such as protein translation. Under such metabolic stress, the bacteria enter dormancy and develop a resistance to the antibiotics [106]. The latency and antibiotic insensitivity in *Mycobacterium tuberculosis* is partially mediated by TA elements through their roles in regulating metabolic stress [107]. Although the overexpression of Lon is shown to dysregulate the TA system in other bacteria, *Mycobacterium tuberculosis* relies upon two different cytosolic AAA + proteases for the same reason: ClpC1P1P2 and ClpXP1P2 [108]. *Escherichia coli* Lon degrades the RelB antitoxin of the relBE system [18], and during amino acid starvation, the MazE antitoxin of the mazEF system. Additionally, the overproduction of Lon in *Escherichia coli* is detrimental to the yefM-yoeB TA system [34]. The overproduction of Lon leads to the rapid and severe inhibition of antitoxin YefM, which activates the toxin YoeB, ultimately leading to bacterial cell death.

### 4.3. Polyphosphate-Lon Protease Complex during Amino Acid Starvation

In eukaryotes, the intracellular protein degradation in response to amino acid starvation and cellular differentiation is carried out by the ubiquitin conjugation system, wherein an increased E3 ubiquitin ligase activity level and/or decreased deubiquitinase activity level results in the accumulation of a subset of ubiquitinated substrates [109,110]. In *Escherichia coli*, amino acid starvation leads to the accumulation of polyphosphate (polyP), a linear polymer consisting of many hundreds of orthophosphate residues. The accumulation of polyP appears to trigger protein degradation, where polyP forms a complex with the ATP-dependent Lon protease. The formation of this complex aids in the degradation of free ribosomal proteins. The polyP-Lon complex or stringent protease formation is crucial for protein degradation at the onset of starvation. During this process, polyP acts as an adaptor molecule to stimulate Lon-substrate complex formation and does not act as a tag for Lon-mediated degradation [111,112].

## 5. Conclusions

The ATP-dependent Lon protease is ubiquitous and found in most forms of life. As a global regulator, Lon protease controls a wide variety of functions related to bacterial survival and pathogenesis. Apart from regulating the virulence-associated genes in bacteria, Lon modulates the cytokine expression in the host’s cells to facilitate bacterial internalization and dissemination. The ability of Lon to alter the size of its exit pore depending on the presence or absence of oxygen makes it a unique intracellular protease. Lon is important for the stress responses in an array of bacteria. It helps the pathogens to circumvent all forms of stress including heat, oxidative and metabolic stresses. Lon-mediated proteolytic cleavage can either commence upon encountering a degron-tagged polypeptide or by stringent protease formation depending on the environment and nutritional availability.

The ability of Lon to modulate antibiotic resistance and toxin-antitoxin elements indicates that the Lon protease might be a good therapeutic target to address the recalcitrant infections caused by pathogenic bacteria. Additionally, this global regulator is essential for drug tolerance and resistance, thus modulating the evolutionary dynamics of antimicrobial resistance. The species-specific substrate selectivity for Lon will help us to uncover a vista of therapeutic targets against disease-causing pathogens that will be non-toxic to mammalian cells.

## Figures and Tables

**Figure 1 ijms-24-03422-f001:**
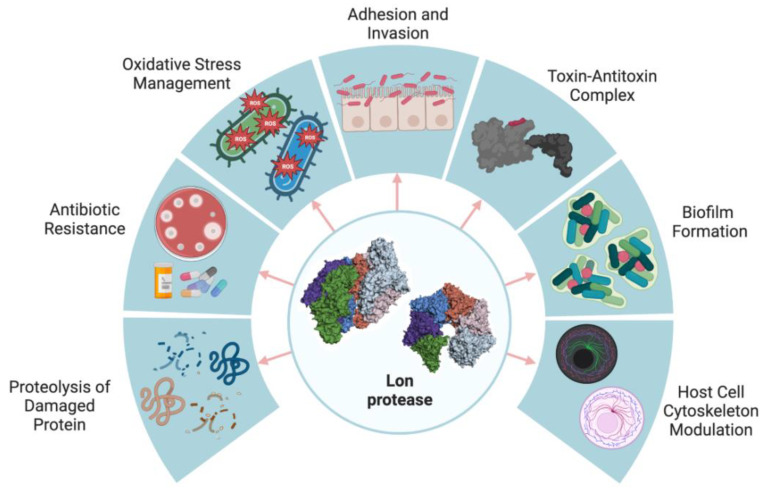
Lon protease regulates various functions in bacteria. Here, the hexamer form of the Lon protease is shown. The figure was created with BioRender.com.

**Figure 2 ijms-24-03422-f002:**
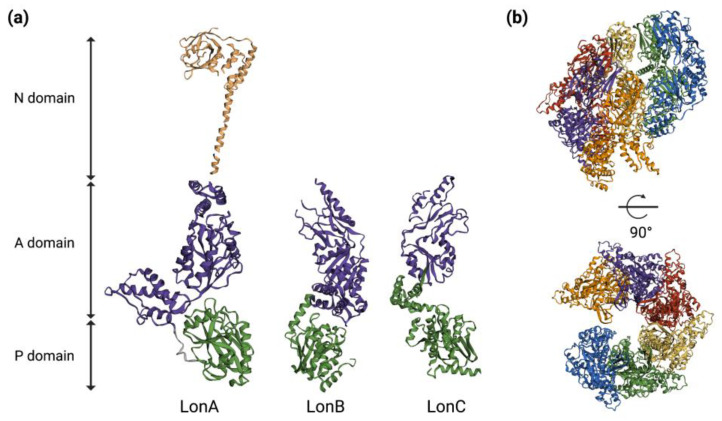
Three-dimensional structures of Lon protease. (**a**) Representative 3D models of single protomers of LonA (PDB ID: 6u5z & 3ljc) of *E. coli*, LonB of *Thermococcus onnurineus* (PDB ID: 3k1j) and LonC of *Meiothermus taiwanensis* (PDB ID: 4fw9). (**b**) Cryo-EM structure of LonA in *E. coli* (PDB ID: 6u5z) without a substrate. Six different colors represent six chains that form the hexamer. The 3D structures and PDB IDs were adapted from Wlodawer et. al., 2022 [38]. P = Protease; A = ATPase; N = N-terminal.

**Table 1 ijms-24-03422-t001:** Important substrates of bacterial Lon protease.

Bacteria	Substrate(s)	Function(s)
*Escherichia coli* Lon	HupB (DNA-binding protein HU-beta)	Histone-like DNA-binding protein that wraps around the DNA to stabilize it and prevent its denaturation under extreme environmental conditions [56].
	IbpA (Small heat shock protein)	Together with IbpB, it protects the aggregated proteins from irreversible denaturation and extensive proteolysis during heat shock and oxidative stress [57].
	RcsA (Transcriptional regulatory protein)	Component of the Rcs signaling system, which controls the transcription of an array of genes. RcsA binds RcsB to the RcsAB box to regulate gene expression [58].
	RpsB (30S ribosomal protein S2)	Required for ribosomal protein S1 to bind to the 30S subunit [59].
	SoxS (Regulatory protein)	Transcriptional activator of the superoxide response regulon [60]. It also facilitates binding of RNA polymerase to the micF and the nfo promoters [61].
	SulA (Cell division inhibitor)	Component of the SOS system and an inhibitor of cell division [62].
	UmuD	Involved in UV protection and mutation. Poorly processive, error-prone DNA polymerase involved in translesion repair [63].
	TrfA (Plasmid replication initiator protein)	Required for the initiation of plasmid DNA replication, along with host-derived DnaA and other host proteins [64].
	RepE (Replication initiation protein)	Replication initiator in the monomeric form, and autogenous repressor in the dimeric form [65].
	CcdA (antitoxin)	Antitoxin component of a type II toxin-antitoxin (TA) system which inhibits the post-segregational killing (PSK) of plasmid-free cells [66].
*Bacillus subtilis* Lon	SwrA	Modulator of the two-component system DegSU; it is important for swarm motility [67].
*Yersinia pestis* Lon	YmoA (*Yersinia* modulator A)	Small histone-like protein which is required for *Yersinia* T3SS induction [68].
	RsuA (Ribosomal small subunit pseudouridine synthase A)	Responsible for synthesis of pseudouridine from uracil-516 in 16S ribosomal RNA [69].
	Fur (Ferric uptake regulation protein)	Acts as a repressor, employing Fe^2+^ as a cofactor to bind the operator of the iron transport operon [70].
*Caulobacter crescentus* Lon	Cell-Cycle-Regulated DNA Methyltransferase (CcrM)	Regulates the methylation of chromosomal DNA and cellular differentiation [54].

**Table 2 ijms-24-03422-t002:** Virulence-associated genes regulated by Lon protease in bacterial pathogens.

Bacteria	Virulence-Associated Genes
*Pseudomonas aeruginosa*	Type III secretion system genes: *pscF* (needle complex), *popB* and *popD* (translocation apparatus), *exsC* (regulatory protein) and *exoS* (effector protein cytotoxin).
*Salmonella enterica* serovar Typhimurium	*Salmonella* pathogenicity island 1 (SPI-1) genes: *hilA* (transcriptional regulator of SPI-1), *invF* (necessary for the activation of *sigDE*, *sicAsipBCDA* and *sopE*), *sipA* (potentiates *SipC* activity; alters host cell actin cytoskeleton) and *sipC* (interferes with host cytoskeleton and enables efficient bacterial internalization)
*Yersinia* species	*Yersinia* outer proteins (Yops) and type III secretion system (T3SS).
*Escherichia coli*	Locus of Enterocyte Effacement (LEE) pathogenicity island associated virulence genes: *ler* (LEE-encoded regulator) and *grlA* (Type III secretion system LEE transcriptional regulator GrlA) and *espA* (Type III secretion system LEE translocon filament protein EspA) and *nleA* (Type III secretion system effector NleA) and *marA* (Multiple antibiotic resistance protein MarA).

## Data Availability

Data sharing is not applicable.

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
