# Peer review of "Structure, Substrate Specificity and Role of Lon Protease in Bacterial Pathogenesis and Survival"

_ijms, 2023, doi:10.3390/ijms24043422_

Round 1

Reviewer 1 Report

Comments on the manuscript “Structure, substrate specificity and role of Lon protease in bacterial pathogenesis and survival.”

A short overview on the protease Lon is given. Since there is a large number of publications where the different properties of Lon are described it is certainly a difficult task to select the major aspects. Information is given in a very concentrated form – the reviewer missed some points in the first reading. Although I am not an expert in proteases I missed some items which I found to be interesting: What is the size of the protein degradation products – peptides or amino acids? Lon interaction with polyphosphates is not mentioned.

Fig.1: You could mention in the legend that the dodecamer of Lon is shown.

Line 52: possibly better, ….. persister formation ……

Line 89: .... are reported....

Line 96 – 100: Difficult to understand –the head to head oligomerization is only discussed as a possibility in the reference (38)

Line 113: Why do you not use HupB as usual in E. coli instead of HUß

Line 137: Give the reference number for Nishi et al. [4]

Line 232: Secretion of H2O2 and outer membrane proteins – give reference

Reviewer 2 Report

The manuscript entitled "Structure, substrate specificity and role of Lon protease in bacterial pathogenesis and survival" by Kirthika et al. is an interesting read. However, I would like to make some general suggestions to improve the manuscript:

1. I think it would be good to summarise a range of substrates of Lon protease in bacteria in a tabular format. Additionally, it would be more informative and easier for the readers if the virulence proteins upregulated by the Lon-protease mediated proteolysis were summarised in a table. Furthermore, the inclusion of other pathogenic bacteria, such as Escherichia coli, and Brucella abortus in section 3 would be more advantageous.

2. Adding some representative 3D structures of Lon protease family members that were previously solved would increase the quality of the manuscript. Section 2.1 is seriously missing them. Also, I was wondering whether different Lon variants have any unique preference in recognition and cleavage sites of their shared substrates. What keeps different substrates unique, is there any structural factor involved?

3. Conclusion is very weak at the moment and does not represent this review's content and main aim. Please elucidate their structural and substrate diversity and how to harness them for the potential therapeutic target.

3. Lastly, on a simple note, please be consistent with the bacterial genus and species throughout the manuscript with italics. The author list also ends with "and". 

Thank you

Round 2

Reviewer 2 Report

Thank you for the revision. The updated manuscript reads well.

Only a minor suggestion: After writing the complete name of a bacteria such as Escherichia coli in the first mention, the genus name can be shortened to just the capital letter (E. coli).